# Application of Central Composite Design for Optimization of Adsorption of Chromium(VI) by *Spirulina platensis* Algae Biomass

Santiago Urréjola-Madriñán [1,*], Iñaki Paz-Armada [1], Claudio Cameselle [2] and Susana Gouveia [2]

1   Defense University Center at the Spanish Naval Academy, 36920 Marin, Spain
2   BiotecnIA Group, Chemical Engineering, Universidade de Vigo, 36310 Vigo, Spain
*   Correspondence: urrejola@cud.uvigo.es

**Abstract:** Algal biomass from *Spirulina platensis* has been tested for the adsorption of chromium (VI) in aqueous effluents. The study was conducted using a central composite experimental design. The selected variables were: biomass (0.25–0.75 mg), initial chromium concentration (100–500 mg/L), and contact time (3–8 h). This study proved that spirulina biomass shows good adsorption capacity in the experimental space selected for the central composite experimental design (CCD). The maximum adsorption capacity was 40 mg Cr/g of biomass in the tests with 500 mg/L of Cr(VI) and 0.25 g of spirulina. The statistical analysis confirmed that the adsorption capacity can be modelled using a linear equation that only depends on the initial chromium concentration and the biomass dose. These results suggest that the adsorption of Cr in spirulina raw biomass can be considered in the development of large-scale applications.

**Keywords:** *Spirulina platensis* algae; chromium; adsorption; central composite design

## 1. Introduction

Chromium is a very toxic metal for living organisms and humans [1]. The use of chromium in various industrial activities (pigments, electroplating, stainless steel, etc.) [2] results in the release of chromium into the environment, with subsequent toxic effects [3] on ecosystems. The salts of Cr(VI) are of particular environmental concern because chromate and dichromate are very soluble, movable and bioavailable [4]. Many resources have been dedicated to developing various practical and cost-effective methods for the removal of Cr(VI) from water and wastewater [5]. In the recent literature, various methods have been proposed and tested for the removal of Cr(VI) from water and wastewater [6]. These methods include: adsorption [7], membrane filtration [8], ion exchange [9], and electrochemical treatment methods [10]. Among these technologies, adsorption with an inexpensive biomass and biomass-derived materials were preferred because of their availability, low cost, ease of large-scale implementation and good removal capacity [11–13].

Various studies in the literature have tested the capacity of biomass and agro-forestry wastes for the adsorption of Cr(VI) [14]. The adsorption capacities for biomass or derived materials (e.g., activated carbon) ranged from 4 mg Cr/g of adsorbent to 200 mg/g [15], depending on the chemical nature of the adsorbent and the experimental adsorption conditions. The modification of the biomass with physicochemical and/or thermal pretreatments was also used to enhance the adsorption capacity [16]. However, modification of the biomass or any other pretreatment increased the energy and chemical costs, as well as adding complexity to the scale-up of the adsorption process. Therefore, it is preferable to use raw biomass, mainly from waste biomass sources, to develop cost-effective adsorption processes [13].

Algae biomass has been reported to show good adsorption capacity for metals and other organic contaminants [17]. Algae biomass is a common waste in coastal areas with no

specific destiny or applications, so it is a good candidate as a low-cost, adsorbent material for the retention of metals in water and wastewater. Microalgae can be cultured in natural or synthetic media with very low cost and energy requirements. For example, *Spirulina platensis* can be cultured in wastewater as a fundamental stage in the effluent treatment and nutrient removal [18]. The produced biomass can be separated and used as an adsorbent for metals [19].

In this study, we selected the biomass of the algae *Spirulina platensis* to test its capacity for the adsorption of Cr(VI) in aqueous effluents. The study used a central composite experimental design to identify the influence of chromium concentration, solid–liquid ratio (biomass-Cr solution) and contact time on the adsorption process. Spirulina biomass was selected because the algae species grows very fast in photo-bioreactors or raceways with low nutrient requirements [20]. The final objective of this study was to develop a practical adsorption technology for Cr(VI) in waste effluents.

## 2. Materials and Methods

### 2.1. Algae Biomass

The algal biomass from *Spirulina platensis* has been selected as a residual biomass source for the adsorption of chromium (VI) in aqueous solutions. The biomass of spirulina was obtained from a commercial produce commonly used for human or animal food (Celtalga, Santiago, Spain). Spirulina biomass contains about 70% protein. It also contains a significant amount of lipids, vitamins (mainly B12 and β-carotene), and various pigments (e.g., xanthophylls and chlorophylls). Table 1 shows the typical composition of the commercial spirulina biomass [21].

**Table 1.** Typical composition of dry spirulina biomass.

| Component | Composition |
|---|---|
| Phycocyanin | 140 g/kg |
| Chlorophyll | 6.1–10 g/kg |
| Carotenoid | 3.7 g/kg |
| β-carotene | 1.5–1.9 g/kg |
| Protein | 55–70% |
| Moisture | 4–7% |
| Ashes * | 6–13% |

Note(s): * Ashes: inorganic residue after ignition at 550 °C for 1 h.

### 2.2. Chromium Solution

The solution of Cr(VI) was prepared by weighting the necessary amount of potassium dichromate, of analytical grade, in de-ionized water for a final concentration of 1000 mg/L of Cr. This solution was stored at room temperature in the dark, and used as stock solution to prepare chromium solutions in the adsorption tests with spirulina biomass.

### 2.3. Batch Adsorption Tests

The adsorption of Cr in spirulina biomass was studied using a central composite experimental design with three variables: dose of algae biomass, initial Cr concentration in the liquid, and contact time between the biomass and Cr(VI) solution. Each adsorption test was conducted by mixing specific amounts of biomass and 50 mL of Cr solution in 100 mL plastic bottles. The mixture was shaken in a rotatory shaker at 250 rpm for a maximum of 10 h. The temperature in the shaker was controlled at $20 \pm 1$ °C. At the end of the tests, the biomass and the supernatant liquid were separated using a microfiber glass filter. The residual Cr concentration in the liquid was determined by a colorimetric method [22]. The percentage of removed Cr from the solution and adsorbed in the biomass was calculated considering the initial ($Cr_{initial}$) and final ($Cr_{final}$) Cr concentrations in each test as shown in Equation (1). The adsorption capacity (q) of Cr in the biomass was calculated with Equation (2) considering the initial and final Cr concentrations in the liquid phase, the

volume of the liquid phase (50 mL) and the amount of biomass used in each test (biomass). The sorption capacity was expressed as mg/g (mg of Cr per g of biomass).

$$\text{Adsorbed Cr (\%)} = (\text{Cr}_{\text{initial}} - \text{Cr}_{\text{final}})/\text{Cr}_{\text{initial}} \tag{1}$$

$$q \text{ (mg/g)} = (\text{Cr}_{\text{initial}} \text{ (mg/L)} - \text{Cr}_{\text{final}} \text{ (mg/L)}) \times 50 \times \times 10^{-3} \text{ mL/Biomass (g)} \tag{2}$$

### 2.4. Determination of Cr

The concentration of Cr(VI) in the liquid solutions before and after the adsorption tests was determined using the diphenylcarbazide colorimetric method [22]. Cr in solution reacts with diphenylcarbazide, forming a colored compound that was determined by absorbance in a UV-Vis spectrophotometer at 540 nm. The analytical procedure used 4 mL of sample in a test tube. Firstly, 0.5 mL of sulfuric acid 20 mg/L was added to the sample. Then, 0.5 mL of diphenylcarbazide 1 g/L was added and the test tube was thoroughly mixed. The colored complex between Cr(VI) and diphenylcarbazide was formed. After 5 min, the color of the complex was completely developed and the absorbance was measured at 540 nm, using a blank prepared with deionized water as reference. The concentration of Cr(VI) was determined by comparing the absorbance of the sample with the absorbance of standard solutions in the range 0.5–2 mg/L of Cr(VI).

### 2.5. Central Composite Desing

A central composite experimental design (CCD) was used to determine the influence of the three selected variables in the adsorption of Cr(VI) in the spirulina biomass. The CCD with real and coded variables is described in Table 2. The CCD includes 20 experiments, carried out at specific levels of the three variables, as shown in Table 3. The repeated tests at the central point (0, 0, 0) for the three variables (Tests 15–20 in Table 3) were used to evaluate the experimental error associated with the experimental adsorption procedure and analysis. Specific details about the use and applications of CCD and its ANOVA analysis can be found at Leardi [23]. The mathematical treatment and calculations of the results of the CCD and the statistical analysis ANOVA were completed with Expert-Design software (STAT-EASE, Minneapolis, MN, USA).

**Table 2.** Central composite design for the adsorption of Cr in spirulina biomass.

| Variable | Factor | Factor Level and Variable Value | | | | | Units |
|---|---|---|---|---|---|---|---|
| | | $-\alpha$ | $-1$ | $0$ | $1$ | $+\alpha$ | |
| Algal biomass | A | 0.0795 | 0.25 | 0.5 | 0.75 | 0.9205 | g |
| Cr(VI) | B | 97.7 | 200 | 350 | 500 | 602.3 | mg/L |
| Time | C | 1.295 | 3 | 5.5 | 8 | 9.705 | h |

**Table 3.** The 20 adsorption tests of the central composite experimental design.

| Test | Factor A | Factor B | Factor C | Biomass | Cr(VI) | Time | pH | EC * |
|------|----------|----------|----------|---------|--------|------|-----|------|
|      |          |          |          | (g)     | (mg/L) | (h)  |     | (mS/cm) |
| 1    | −1       | −1       | −1       | 0.25    | 200    | 3    | 5.94 | 0.96 |
| 2    | 1        | −1       | −1       | 0.75    | 200    | 3    | 6.88 | 1.7  |
| 3    | −1       | 1        | −1       | 0.25    | 500    | 3    | 5.71 | 1.64 |
| 4    | 1        | 1        | −1       | 0.75    | 500    | 3    | 6.41 | 2.35 |
| 5    | −1       | −1       | 1        | 0.25    | 200    | 8    | 6.31 | 1    |
| 6    | 1        | −1       | 1        | 0.75    | 200    | 8    | 6.06 | 0.97 |
| 7    | −1       | 1        | 1        | 0.25    | 500    | 8    | 5.34 | 1.39 |
| 8    | 1        | 1        | 1        | 0.75    | 500    | 8    | 5.61 | 1.55 |
| 9    | −1.682   | 0        | 0        | 0.0795  | 350    | 5.5  | 5.57 | 0.98 |
| 10   | 1.682    | 0        | 0        | 0.9205  | 350    | 5.5  | 5.67 | 1.03 |
| 11   | 0        | −1.682   | 0        | 0.5     | 97.7   | 5.5  | 6.74 | 1.05 |
| 12   | 0        | 1.682    | 0        | 0.5     | 602.3  | 5.5  | 5.81 | 1.59 |
| 13   | 0        | 0        | −1.682   | 0.5     | 350    | 1.295 | 6.21 | 1.33 |
| 14   | 0        | 0        | 1.682    | 0.5     | 350    | 9.705 | 6.08 | 1.5  |
| 15   | 0        | 0        | 0        | 0.5     | 350    | 5.5  | 5.89 | 1.16 |
| 16   | 0        | 0        | 0        | 0.5     | 350    | 5.5  | 5.89 | 1.16 |
| 17   | 0        | 0        | 0        | 0.5     | 350    | 5.5  | 5.89 | 1.16 |
| 18   | 0        | 0        | 0        | 0.5     | 350    | 5.5  | 5.89 | 1.16 |
| 19   | 0        | 0        | 0        | 0.5     | 350    | 5.5  | 5.89 | 1.16 |
| 20   | 0        | 0        | 0        | 0.5     | 350    | 5.5  | 5.89 | 1.16 |

Note(s): * EC: Electrical conductivity.

## 3. Results

The adsorption of Cr(VI) in spirulina biomass is shown in Table 4 for the 20 CCD experiments. At the end of the tests, the residual amount of Cr in the solution was measured and the adsorbed amount of Cr was calculated using Equations (1) and (2). The absorbed amount of Cr is reported in the last two columns of Table 4 as percentages and adsorption capacity (q).

**Table 4.** Results of the adsorption of Cr(VI) in spirulina biomass using a central composite design.

| Test | Factor A | Factor B | Factor C | Residual Cr (mg/L) | Adsorbed Cr (%) | Adsorbed Cr q (mg/g)) |
|------|----------|----------|----------|--------------------|------------------|------------------------|
| 1    | −1       | −1       | −1       | 175.70 | 12.15 | 17.8 |
| 2    | 1        | −1       | −1       | 152.74 | 23.63 | 9.6  |
| 3    | −1       | 1        | −1       | 285.15 | 42.97 | 40.4 |
| 4    | 1        | 1        | −1       | 228.75 | 54.25 | 21.0 |
| 5    | −1       | −1       | 1        | 130.58 | 34.71 | 17.8 |
| 6    | 1        | −1       | 1        | 128.02 | 35.99 | 10.6 |
| 7    | −1       | 1        | 1        | 288.35 | 42.33 | 40.4 |
| 8    | 1        | 1        | 1        | 282.95 | 43.41 | 21.0 |
| 9    | −1.682   | 0        | 0        | 306.11 | 12.54 | 35.7 |
| 10   | 1.682    | 0        | 0        | 269.14 | 23.10 | 3.1  |
| 11   | 0        | −1.682   | 0        | 57.36  | 41.29 | 2.4  |
| 12   | 0        | 1.682    | 0        | 159.90 | 73.45 | 38.4 |
| 13   | 0        | 0        | −1.682   | 237.35 | 32.19 | 19.4 |
| 14   | 0        | 0        | 1.682    | 202.85 | 42.04 | 20.3 |
| 15   | 0        | 0        | 0        | 146.23 | 58.22 | 18.6 |
| 16   | 0        | 0        | 0        | 147.01 | 57.12 | 18.3 |
| 17   | 0        | 0        | 0        | 149.54 | 55.46 | 19.5 |
| 18   | 0        | 0        | 0        | 144.67 | 59.34 | 20.8 |
| 19   | 0        | 0        | 0        | 145.23 | 59.02 | 19.7 |
| 20   | 0        | 0        | 0        | 146.88 | 58.55 | 19.5 |

### 3.1. Percentage of Adsorbed Chromium

The percentage of adsorbed chromium in the selected experimental space defined in the CCD (Table 4) can be represented with the quadratic model in Equation (3).

$$OF = b_0 + b_1 \times A + b_2 \times B + b_3 \times C + b_{12} \times AB + b_{13} \times AC + b_{23} \times BC + b_{11} \times A^2 + b_{22} \times B^2 + b_{33} \times C^2 \quad (3)$$

where objective function (OF) is the percentage of adsorbed chromium; A, B, and C are the factors of the CCD expressed as coded variables; and $b_{ij}$ is the contribution of each factor to the value of the objective function. Equation 4 shows the fitting of the adsorbed Cr results to the quadratic model using Design-Expert software. The coefficient values show the influence of each factor on the objective function. The sign of the coefficient shows whether the contribution of each factor to the OF is positive or negative. As can be seen, Cr(IV) initial concentration (factor B) is the variable with the higher influence, but the highest influence corresponds to the quadratic biomass $A^2$, with a coefficient of $-14.28$. Contact time had a lower influence in the OF than biomass and chromium concentration. Overall, we can conclude that adsorption percentage is favored at a higher Cr concentration and biomass dose with a low influence of contact time.

$$\text{Adsorbed Cr (\%)} = +58.22 + 3.14 \times A + 9.56 \times B + 2.93 \times C - 0.050 \times AB \\ - 2.55 \times AC - 5.80 \times BC - 14.28 \times A^2 - 0.30 \times B^2 - 7.46 \times C^2 \quad (4)$$

Equation (4) is written in terms of coded variables, but it can be transformed to real variables: biomass in g, Cr in mg/L, and time in h, as shown in Equation (5). This equation can be used to predict the percentage of adsorbed chromium for any test in the experimental space of the CCD.

$$\text{Adsorbed Cr (\%)} = -112.90043 + 263.98172 \times \text{Biomass} + 0.15879 \times \text{Cr} \\ + 21.75256 \times \text{Time} - 1.33333 \times 10^{-3} \times \text{Biomass} \times \text{Cr} - 4.08000 \times \text{Biomass} \times \text{Time} - 0.015467 \times \\ \text{Cr} \times \text{Time} - 228.51260 \times \text{Biomass}^2 - 1.32889 \times 10^{-5} \, \text{Cr}^2 - 0.116 \times \text{Time}^2 \quad (5)$$

The analysis of variance (ANOVA) of the percentage of adsorbed chromium and the quadratic model was calculated using Design-Expert software (Table 5). The ANOVA indicates that the model is statistically significant. The quadratic model represents the percentage of adsorbed Cr with a confidence level higher than 90%. The ANOVA can also identify the significant and non-significant variables in the model. The variables with a probability below 0.1 (Table 5, *p*-value) are significant, whereas the variables with a probability higher than 0.1 are not significant for the model. The non-significant variables can be removed from the model (Equations (4) and (5)). The significant variables are B, $A^2$ and $C^2$: the initial concentration of Cr(VI), the square of the biomass, and the square of contact time. The non-significant variables can be removed from the model equation, resulting a much simpler equation (Equations (6) and (7)).

$$\text{Adsorbed Cr (\%)} = b_0 + b_2 \times B + b_{11} \times A^2 + b_{33} \times C^2 \quad (6)$$

$$\text{Adsorbed Cr (\%)} = -112.90043 + 0.15879 \times \text{Cr} - 228.51260 \times \text{Biomass}^2 - 0.116 \times \text{Time}^2 \quad (7)$$

**Table 5.** ANOVA of percentage of adsorbed Cr in biomass with the quadratic model of Equation (4).

| Source | Sum of Squares | Degrees of Freedom | Mean Square | F Value | *p*-Value Prob > F | |
|---|---|---|---|---|---|---|
| Model | 5333.44 | 9 | 592.60 | 2.88 | 0.0575 | significant |
| A-Biomass | 134.7 | 1 | 134.70 | 0.65 | 0.4374 | |
| B-Cr(VI) | 1248.99 | 1 | 1248.99 | 6.07 | 0.0335 | |
| C-Time | 117.31 | 1 | 117.31 | 0.57 | 0.4677 | |
| AB | 0.020 | 1 | 0.020 | $9.717 \times 10^5$ | 0.9923 | |
| AC | 52.02 | 1 | 52.02 | 0.25 | 0.6260 | |
| BC | 269.12 | 1 | 269.12 | 1.31 | 0.2795 | |
| $A^2$ | 2939.57 | 1 | 2939.57 | 14.28 | 0.0036 | |
| $B^2$ | 1.29 | 1 | 1.29 | $6.259 \times 10^{-3}$ | 0.9385 | |
| $C^2$ | 801.68 | 1 | 801.68 | 3.89 | 0.0767 | |
| Residual | 2058.32 | 10 | 205.83 | | | |
| Lack of Fit | 2058.32 | 5 | 411.86 | | | |
| Pure Error | 0.000 | 5 | 0.000 | | | |
| Cor Total | 7391.76 | 19 | | | | |

The model (Equation (7)) can be used to predict the adsorption percentage of Cr(VI) in spirulina biomass in the experimental space selected in the CCD. The adsorption percentage depends on the biomass dose, the initial Cr(VI) concentration and the contact time. The graphical representation of the Equation (6) is shown in Figure 1. As can be seen, the adsorption percentage increases with the initial concentration of Cr(VI) and intermediate biomass values (Figure 1a). The adsorption percentage is also favored for a 5 h contact time with 0.5 g of biomass (Figure 1b). Finally, Figure 1c shows that the adsorption percentage is favored at higher Cr(VI) concentrations and intermediate (about 5 h) values of contact time.

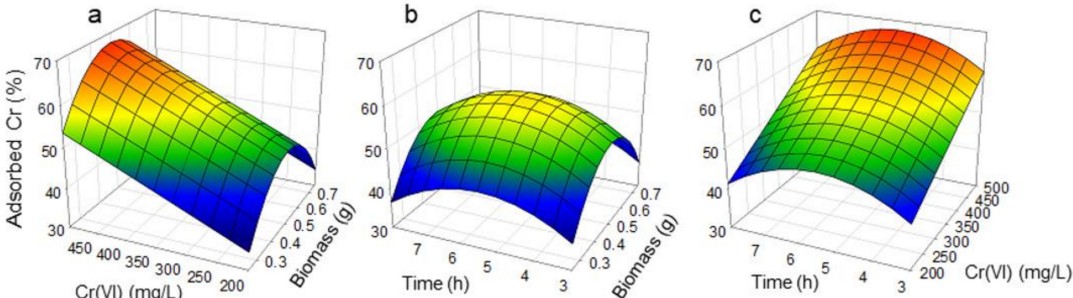

**Figure 1.** Surface response of the quadratic model in Equation (7) for the percentage of Cr adsorbed in the spirulina biomass versus (**a**) Cr concentration and biomass dose, (**b**) contact time and biomass dose, and (**c**) contact time and Cr concentration.

### 3.2. Sorption Capacity

The sorption capacity was determined from the residual concentration of Cr in the liquid phase at the end of the tests (Table 4). The sorption capacity, q, was calculated as the amount of adsorbed Cr (in mg) in 1 g of spirulina biomass. The experimental results were fitted to the linear model in Equation (8) using the Design-Expert software.

$$OF \text{ (sorption capacity, mg/g)} = b_0 + b_1 \times A + b_2 \times B + b_3 \times C \tag{8}$$

The ANOVA analysis of the model is shown in Table 6. The low *p*-value confirms that the model is significant to represent the sorption capacity as an objective function for a confidence level of 99%. The variable "time" is not significant (high *p*-value) and the other two variables, biomass and Cr(VI) initial concentration, are significant for a confidence level of 99% (see *p*-value in Table 6). As a result, the liner model in Equation (8) can be

simplified, as shown in Equation (9) (model for coded variables) and Equation (10) (model for real variables).

$$\text{Sorption capacity (mg/g)} = +19.40 - 9.70 \times A + 11.31 \times B \tag{9}$$

$$\text{Sorption capacity (mg/g)} = +11.29 - 38.79 \times \text{Biomass} + 0.075 \text{ Cr(VI)} \tag{10}$$

**Table 6.** ANOVA of sorption capacity q with the linear model of Equation (8).

| Source | Sum of Squares | Degrees of Freedom | Mean Square | F Value | *p*-Value Prob > F | |
|---|---|---|---|---|---|---|
| Model | 3034.25 | 3 | 1011.42 | 11.60 | 0.0003 | significant |
| A-Biomass | 1284.43 | 1 | 1284.43 | 14.73 | 0.0015 | |
| B-Cr(VI) | 1746.29 | 1 | 1746.29 | 20.03 | 0.0004 | |
| C-Time | 3.53 | 1 | 3.53 | 0.040 | 0.8431 | |
| Residual | 1394.85 | 16 | 87.18 | | | |
| Lack of Fit | 1394.85 | 11 | 126.80 | | | |
| Pure Error | 0.000 | 5 | 0.000 | | | |
| Cor Total | 4429.10 | 19 | | | | |

The graphical representation of the model in Equation (10) (Figure 2) informs that the Cr sorption capacity in spirulina biomass increases with the initial concentration of Cr and lower biomass values. The influence of the contact time on the sorption capacity is irrelevant in the tested experimental range (3–10 h). Considering the mathematical model in Equation (10) and the surface response in Figure 2, the highest sorption capacity was obtained at 500 mg/L of Cr(VI) and 0.25 g of spirulina biomass.

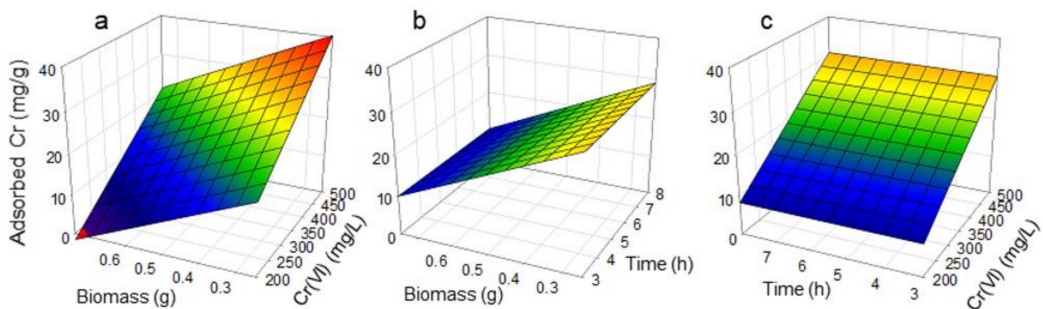

**Figure 2.** Surface response of the linear model in Equation (10) for the sorption capacity q of Cr in spirulina biomass versus (**a**) Cr concentration and biomass dose, (**b**) contact time and biomass dose, and (**c**) contact time and Cr concentration.

## 4. Discussion

The results of this study proved that *Spirulina platensis* biomass shows the appropriate characteristics for the adsorption of chromium from aqueous solutions. The statistical analysis determined that the maximum adsorption capacity (q) is about 40 mg Cr/g biomass under the test conditions of 0.25 mg of biomass and 500 mg/L of initial Cr concentration. This result is slightly higher than previous results reported in the literature for the adsorption of Cr(VI) and Cr(III) in spirulina at similar pH and temperature conditions [24–29]. The adsorption capacity of spirulina may be increased with modifications to the biomass or chemical/thermal pretreatments. The adsorption capacity is also affected by the physiochemical conditions of the system. For example, higher temperatures (60 °C) or low pH tend to increase the adsorption capacity by up to 200 mg/g [25,28]. However, acidification of effluents is not always feasible due to the environmental impact of acid effluents after adsorption; the final effluent may require neutralization. The increase in temperature is going to increase the energy costs to unacceptable levels. The physicochemical pretreatment

of biomass to increase the adsorption capacity is not going to compensate the cost in chemicals and energy [30]. Furthermore, the physicochemical pretreatment of biomass may be feasible at lab scale, but it will be very costly and complex for large-scale applications [31].

The results in this study suggest it is possible to develop a practical adsorption process of Cr(VI) with spirulina biomass. Table 7 shows the literature reports on the use of spirulina biomass in the adsorption of chromium from aqueous effluents. As can be seen, the raw biomass directly obtained from bioreactors or raceway cultures can be used in the treatment of Cr-contaminated effluents. Some modifications to the biomass (e.g., methylation) or the use of wasted biomass from various processes (e.g., biodiesel production of pigment extraction) can be used in the adsorption of Cr with a very good adsorption capacity, reaching values as high as 212 mg of Cr/g of adsorbent.

**Table 7.** Adsorption of Cr on spirulina biomass.

| Biomass | Contaminant | Adsorption Conditions | Adsorption Capacity (q) | Ref. |
|---|---|---|---|---|
| *Spirulina sp.* Raw biomass | Cr(VI) | pH 5 1 h | 90.91 mg/g | [24] |
| *Spirulina platensis.* Waste from the biodiesel extraction | Cr(VI) | pH 1 60 °C. 2 h | 45.5–60 mg/g | [25] |
| *Spirulina platensis.* Methylated biomass | Cr(VI) | pH 7–8 | 7.4–16.7 mg/g | [26] |
| *Spirulina platensis.* Fresh biomass | Cr(VI) | pH 1.5 25 °C | 188 mg/g | [27] |
| *Spirulina platensis.* Spent biomass after the extraction of β-carotene | Cr(VI) | pH 1.5 25 °C | 212 mg/g | [28] |
| *Spirulina platensis.* Dry biomass | Cr(III) | pH 6 20 °C | 30–36 mg/g | [29] |

The scale-up of the adsorption process requires the continuous and stable production of biomass with enhanced adsorption capacity [31]. The bioreactor design and culture conditions of spirulina may allow for the continuous production of biomass with stable characteristics. It is important to design a biomass collection system and the transfer of biomass to the adsorption unit. Finally, a safe destiny of the contaminated biomass must be identified and implemented. As an alternative, the culture of spirulina in the Cr-contaminated effluents [29] is an interesting option to combine the production of biomass and the adsorption process.

## 5. Conclusions

This study proved that spirulina biomass showed a good adsorption capacity in the experimental space selected for the CCD. The maximum adsorption capacity was 40 mg Cr/g of biomass in the tests with 500 mg/L of Cr(VI) and 0.25 g of spirulina. The statistical analysis confirmed that the adsorption capacity can be modelled using a linear equation that only depends on the initial chromium concentration and the biomass dose. The linear model is statistically significant at a 99% level of confidence. A quadratic model was used to modelize the adsorption expressed in terms of percentage in the selected experimental space for the CCD (biomass: 0.25–0.75 g, Cr concentration: 200–500 mg/L, contact time: 3–8 h). The results in this study suggest that spirulina biomass shows interesting perspectives for the development and scaling of the Cr(VI) adsorption process for the treatment of contaminated effluents and wastewater.

**Author Contributions:** Conceptualization, S.U.-M., C.C. and S.G.; methodology, S.U.-M., C.C. and S.G.; software, S.G.; validation, S.U.-M. and C.C.; formal analysis, S.G.; investigation, I.P.-A.; resources, S.U.-M.; data curation, C.C. and S.G.; writing—original draft preparation, S.U.-M.; writing—review and editing, C.C.; visualization, I.P.-A.; supervision, S.U.-M.; project administration, S.U.-M. and C.C.; funding acquisition, S.U.-M. All authors have read and agreed to the published version of the manuscript.

**Funding:** This research received no external funding.

**Institutional Review Board Statement:** Not applicable.

**Informed Consent Statement:** Not applicable.

**Conflicts of Interest:** The authors declare no conflict of interest.

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
