# Peer review of "Application of Central Composite Design for Optimization of Adsorption of Chromium(VI) by Spirulina platensis Algae Biomass"

_water, doi:10.3390/w14162539_

Round 1

Reviewer 1 Report

What are the originality and novelty?  In the discussion section, the author mentioned the results of this study are comparable with the results in the published literature.  What is the unique feature that differentiates it from other published works? 

What is the improvement from the previous version?

Reviewer 2 Report

Author mentioned in this article about the algal biomass from Spirulina platensis which was tested for the adsorption of chromium (VI). in aqueous effluents. The manuscript reflects some facts about the biomass and its adsorptive properties. However some comments need to be incorporated before publication.

Please elaborate the introduction section by adding one paragraph about this choosen heavy metal e.g Journal of Environmental Chemical Engineering 9 (5), 106111....why author choose Cr? 

Also add one paragraph why choose this biomass e.g. Polymers 14 (4), 845.

Please add some more references to support the work done in this article in all sections.

Add conclusion of the present study as section 5.

please mention the relevant reference to support the statement specially in results and discussion section.

In discussion section, please add some critical text to support the obtained results.

Reviewer 3 Report

See my comments in the attached word file.

Round 2

Reviewer 1 Report

The authors of the manuscript "Application of central composite design for optimization of adsorption of chromium (VI) by Spirulina platensis algae biomass" has made some improvements from previous version.

However, the article needs some improvements from the current version. 

1.     There are a few typing mistakes as well, and authors are advised to proofread the text carefully. For example: typo in line 44, “spirulina platensis” in lines 48 and 211needs to be italic.

2.     Redundant in formation in lines 55-56. 

3.     Please specify the company information. 

4.     Table 1: Ashes 6-13%? Is there any other component in the spirulina biomass? 

5.     Define “EC” in table 3

Author Response

  1. There are a few typing mistakes as well, and authors are advised to proofread the text carefully. For example: typo in line 44, “spirulina platensis” in lines 48 and 211needs to be italic.

Revised and corrected throughout the paper. Changes were highlighted in red.

  1. Redundant in formation in lines 55-56.

Redundant information was deleted from sentences in lines 54-58. Some sentences were rewritten for clarity.

  1. Please specify the company information.

Biomass was obtained from the spin-off company CELTALGA, Santiago de Compostela, Spain.

  1. Table 1: Ashes 6-13%? Is there any other component in the spirulina biomass?

Ash refers to the inorganic residue remaining after ignition, at 550ºC for 1 hour, of organic matter in a biomass sample. This information is included in the manuscript.

  1. Define “EC” in table 3

EC = Electrical conductivity. It is included in the manuscript.

Reviewer 2 Report

Can be accept

Author Response

Thank you for your time and efforts revising and commenting our manuscript

Reviewer 3 Report

The authors have worked on the reviewer's comments and provided the necessary revisions.

Author Response

Thank you for your time and efforts revising and commenting our manuscript

This manuscript is a resubmission of an earlier submission. The following is a list of the peer review reports and author responses from that submission.

Round 1

Reviewer 1 Report

The authors of the manuscript investigated the adsorption of chromium by spirulina. The introduction does not provide sufficient background. I recommend rejecting the manuscript due to low originality and novelty. 

Reviewer 2 Report

The authors of the manuscript " Adsorption of chromium by spirulina using a central composite design " presents a very interesting study in water research. The article is written in a clear and understandable way. The selection of literature is appropriate and logical, there are no unnecessary citations.

However, the article presents serious flaws that need to be addressed:

1: What is the novel point of this study?  In the discussion section, the author mentioned the results in this study, and other results in the literature suggest that it is possible to develop a practical adsorption process.  What is the unique feature that differentiates from others? 

2. Any comparisons between spirulina and other adsorbents such as activated carbon, zeolites, and other algae? 

3. The figures are not well made.  

Reviewer 3 Report

I would suggest acceptance of the submitted manuscript.